



# Real-time optimization of wind farms using modifier adaptation and machine learning

Leif Erik Andersson[1] and Lars Imsland[1]

[1]Norwegian University of Science and Technology, Department of Engineering Cybernetics, 7491 Trondheim, Norway

**Correspondence:** Leif Erik Andersson (leif.e.andersson@ntnu.no)

**Abstract.** Real-time optimization (RTO) covers a family of optimization methods that incorporate process measurements in the optimization to drive the real process (plant) to optimal performance while guaranteeing constraint satisfaction. Modifier Adaptation (MA) introduces zeroth and first-order correction terms (bias and gradients) for the cost and constraint functions. Instead of updating the plant model, in MA the optimization problem is updated directly from data guaranteeing to meet the

necessary condition of optimality upon convergence.

The main burden of the MA approach is the estimation of the first-order modifiers of the cost and constraint functions at each RTO iteration. Finite-difference approximation is the most common approach that requires at least nu + 1 steady-state operation points to estimate the gradients, where nu is the number of control inputs. Obtaining these can require a long convergence time. For this reason, this work considers the use of Gaussian process (GP) regression to estimate the plant-model mismatch based

on plant measurements, and replace the usual modifiers by these high order regression functions. GP is a probabilistic, non-parametric modelling technique well known in the machine learning community. The approach is tested on several numerical test cases simulating wind farms. It is shown that the approach is able to correct the model and converges to the plant optimal point. Several improvements for large inputs spaces, which is a challenging problem for the approach presented in the article, are discussed.

## 1    Introduction

Currently the wind turbines in a wind farm are operated at their individual optimal operating point. This control strategy is called *greedy* wind farm control since the interactions between turbines are not taken into account. However, it is expected that the greedy control strategy leads to sub-optimal performance of the wind farm (Steinbuch et al., 1988; Johnson and Thomas, 2009; Barthelmie et al., 2009). A coordinated wind farm controller, which takes the wake interactions between turbines in a

wind farm into account, may result in a superior performance compared to the greedy wind farm controller. The two main wind farm control strategies are axial induction control, e.g. Steinbuch et al. (1988); Corten and Schaak (2003); Horvat et al. (2012); Rotea (2014); Munters and Meyers (2016) and wake steering control, e.g. Medici (2005); Adaramola and Krogstad (2011); Wagenaar et al. (2012); Park et al. (2013); Gebraad and Van Wingerden (2014). The idea behind the former is to deviate the blade pitch and generator torque of the upwind turbine from the greedy control settings. As a consequence, the velocity deficit

in the wake behind the turbine and the power production of the downwind turbine changes. The target net effect is an overall





increase of the power production and possibly an decrease of fatigue loads. However, recent studies suggest that axial induction control using steady-state models to calculate the optimal control settings may be unable to improve the power production of a wind farm (Schepers and Van der Pijl, 2007; Campagnolo et al., 2016; Bartl and Sætran, 2016; Annoni et al., 2016).

The currently more promising wind farm control strategy using steady-state models is wake steering. The goal of wake steering
is to deflect the wake away from the downwind turbine by using the yaw settings of the upwind turbine. Field experiments showing promising results were conducted by Fleming et al. (2017, 2019); Howland et al. (2019). In these experiments lookup tables with optimal yaw settings of each turbine are created with help of an steady-state model. Hence the wind farm is operated in an open-loop control setting.

The steady-state wake models used in model-based control are usually relatively simple. They estimate the velocity deficit in
wakes. For a long time one of the most popular wake models was the Jensen Park model (Jensen, 1983; Katic et al., 1987). Jiménez et al. (2010) developed one of the first steady-state wake models that described wake deflection due to yaw. A recent wake model, which is also used in this study, was presented by Bastankhah and Porté-Agel (2016). It is based on mass and momentum conservation and assumes a Gaussian distribution of the velocity deficit in the wake. The steady-state wake models are able to describe the general behaviour of the wake (Barthelmie et al., 2013; Annoni et al., 2014). Nevertheless, they are just
vague approximations of a complex phenomena that is, in fact, not well understood (Veers et al., 2019). Hence, real time optimization (RTO), which incorporates plant measurements to improve the performance of the wind farm controller, is extremely useful for this process.

Probably one of the most intuitive RTO strategies is the "two-step" approach. Here, first the model parameters are updated, and then new control inputs are computed based on the updated model. The two steps refer to the parameter optimization and
control input optimization, which are performed sequentially (Marchetti et al., 2016). However, the two-step approach cannot guarantee plant optimality upon convergence if the model is structurally incorrect (Marchetti et al., 2016). An example that an improved parameterisation of the steady-state wake model was not able to remove the mismatch between a low order model and a high fidelity model of wake is given in Fleming et al. (2018).

In contrast, modifier adaptation (MA) corrects the cost and constraint functions of the optimization problem directly, and
reaches, under suitable assumptions, true plant optimality upon convergence (Marchetti et al., 2009). The bottleneck of the MA approach is the estimation of the gradients of the objective and constraint functions at each RTO iteration. Finite difference approximation is one of the most common approaches that requires $n_u + 1$ steady-state operation points to estimate the gradients, where $n_u$ is the amount of control inputs. These can lead to a long convergence time, especially for processes with high dimensional input spaces. Therefore, in this work Gaussian process (GP) regression is combined with MA (de Avila Fer-
reira et al., 2018; del Rio Chanona et al., 2019). GP is a probabilistic, non-parametric modelling technique well known in the machine learning community (Rasmussen and Williams, 2006). The GP regression model estimates the plant-model mismatch using plant measurements. Then the GP model is used to correct the original optimization problem and by this improve the optimization of the plant inputs.

The article is structured as follows: In Section 2 the optimization problem is formulated and Gaussian process regression
is explained. In Section 3 the modifier adaptation using Gaussian process regression is presented and the numerical turbine





and wake models introduced. The approach is tested numerically on several examples in Section 4. The article ends with a conclusion.

## 2 Problem formulation

The optimization problem of the steady-state plant performance subject to constraints can be formulated as (Marchetti et al., 2016):

$$\mathbf{u}_p^* = \arg\min_{\mathbf{u}} \phi_p(\mathbf{u}, \mathbf{y}_p(\mathbf{u})) \tag{1a}$$

$$s.t. \; G_{p,j}(\mathbf{u}) := g_{p,j}(\mathbf{u}, \mathbf{y}_p(\mathbf{u})) \le 0, \; j = 1, \ldots, n_g, \tag{1b}$$

$$\mathbf{u} \in \mathcal{U}, \tag{1c}$$

where $\mathbf{u} \in \mathbb{R}^{n_u}$ and $\mathbf{y}_p \in \mathbb{R}^{n_y}$ denote the plant input and output variables, respectively; $\mathbf{u} \in \mathbb{R}^{n_u}$ and $\mathbf{y}_p \in \mathbb{R}^{n_y}$ are the input-output pairs of the wind farm; $\phi_p : \mathbb{R}^{n_u} \to \mathbb{R}$ is the cost function to be minimized; $g_{p,j} : \mathbb{R}^{n_u} \times \mathbb{R}^{n_y} \to \mathbb{R}, \; j = 1, \ldots, n_g$, are the inequality constraint functions; and $\mathcal{U} \subseteq \mathbb{R}^{n_u}$ is the control domain, e.g. box constraints on the control inputs. Formulation (1) assumes that $\phi_p$ and $g_{p,j}$ as functions of $\mathbf{u}$, and $\mathbf{y}_p$ are exactly known. However, in any practical application the exact input-output map of the plant is unknown and instead an approximate model of the system is exploited for the optimization:

$$\mathbf{u}^* = \arg\min_{\mathbf{u}} \phi(\mathbf{u}, \mathbf{y}(\mathbf{u})) \tag{2a}$$

$$s.t. \; G_j(\mathbf{u}) := g_j(\mathbf{u}, \mathbf{y}(\mathbf{u})) \le 0, \; j = 1, \ldots, n_g, \tag{2b}$$

$$\mathbf{u} \in \mathcal{U}, \tag{2c}$$

where the quantities $\phi$, $g_j(\mathbf{u}, \mathbf{y}(\mathbf{u}))$, $\mathbf{u}*$, and $G_j$ refer to the inexact model counterparts of the true plant optimization problem in Eq. (1).

RTO takes advantage of the available measurements to compensate for plant-model mismatch and adapt the model-based optimization problem Eq. (2) to reach plant optimality.

The standard MA approach applies first-order correction terms that are added to the cost and constraint functions to match the necessary conditions of optimality upon convergence (Marchetti et al., 2009). Iteratively the following modified optimization problem is solved:

$$\hat{\mathbf{u}}_{k+1}^* = \arg\min_{\mathbf{u}} \phi(\mathbf{u}, \mathbf{y}(\mathbf{u})) + (\boldsymbol{\lambda}_k^\phi)^T \mathbf{u} \tag{3a}$$

$$s.t. \; G_j(\mathbf{u}) + \varepsilon_{j,k} + (\boldsymbol{\lambda}_k^{G_j})^T(\mathbf{u} - \mathbf{u}_k) \le 0, \; j = 1, \ldots, n_g, \tag{3b}$$

$$\mathbf{u} \in \mathcal{U}, \tag{3c}$$



where $\hat{\mathbf{u}}^*_{k+1}$ is the optimal solution at iteration $k+1$, the $\varepsilon_{j,k} \in \mathbb{R}$ are the zeroth-order modifiers for the constraints, and $\boldsymbol{\lambda}^{\phi}_k$ and $\boldsymbol{\lambda}^{G_j}_k$ are the first-order modifiers for the cost and constraints, respectively. The correction terms are given by:

$$\varepsilon_{j,k} := G_{p,j}(\mathbf{u}_k) - G_j(\mathbf{u}_k), \tag{4a}$$

$$(\boldsymbol{\lambda}^{\phi}_k)^T := \frac{\partial \phi_p}{\partial \mathbf{u}}(\mathbf{u}_k) - \frac{\partial \phi}{\partial \mathbf{u}}(\mathbf{u}_k), \tag{4b}$$

$$(\boldsymbol{\lambda}^{G_j}_k)^T := \frac{\partial G_{p,j}}{\partial \mathbf{u}}(\mathbf{u}_k) - \frac{\partial G_j}{\partial \mathbf{u}}(\mathbf{u}_k). \tag{4c}$$

It is recommended to filter the input update $\hat{\mathbf{u}}^*_{k+1}$ to avoid excessive correction and reduce sensitivity to noise (Marchetti et al., 2016):

$$\mathbf{u}_{k+1} = \mathbf{u}_k + \mathbf{L}(\hat{\mathbf{u}}_{k+1} - \mathbf{u}_k), \tag{5}$$

with $\mathbf{L} = \mathrm{diag}(l_1, \ldots, l_{n_u})$, $l_i \in (0,1]$ where $l_i$ may be reduced to help stabilize the iterations.

The MA scheme requires the estimation of the plant gradients at each RTO iteration, which is experimentally expensive and the main bottleneck for MA implementation in practice (Marchetti et al., 2016).

## 2.1 Gaussian processes

In this section we give a brief outline of GP regression for our purposes, for more information refer to Rasmussen and Williams (2006). GP regression aims to identify an unknown function $f : \mathbb{R}^{n_u} \to \mathbb{R}$ from data. Let the noisy observation of $f(\cdot)$ be given by:

$$y_k = f(\mathbf{u}_k) + \nu_k \tag{6}$$

where the value $f(\cdot)$ is perturbed by Gaussian noise $\nu_k$ with zero mean and variance $\sigma^2_\nu$, $\nu_k \sim \mathcal{N}(0, \sigma^2_\nu)$.

We assume $f(\cdot)$ to follow a GP with a zero mean function and the squared-exponential (SE) covariance function. The choice of the mean and covariance functions assume certain smoothness and continuity properties of the underlying function (Snelson and Ghahramani, 2006). The SE covariance function can be expressed as follows:

$$k(\mathbf{u}_i, \mathbf{u}_j) = \sigma^2_f \exp\left( -\frac{1}{2}(\mathbf{u}_i - \mathbf{u}_j)^T \boldsymbol{\Lambda}^{-1}(\mathbf{u}_i - \mathbf{u}_j) \right) \tag{7}$$

where $\sigma^2_f$ is the covariance magnitude and $\boldsymbol{\Lambda} = \mathrm{diag}(\lambda^2_1, \ldots, \lambda^2_{n_u})$ is a scaling matrix.

Assume we are given a training dataset $\mathcal{D} = \{\mathbf{U}, \mathbf{Y}\}$ of size $M$ consisting of $M$ input vectors $\mathbf{U} = [\mathbf{u}_1, \ldots, \mathbf{u}_M]^\mathsf{T}$ and corresponding observations $\mathbf{y} = [y_1, \ldots, y_M]^\mathsf{T}$ according to Eq. (6). From the GP distribution the data then follows a joint multivariate Gaussian distribution, which can be stated as:

$$p(\mathbf{y}|\mathbf{U}) = \mathcal{N}(\mathbf{0}, \mathbf{K} + \sigma^2_\nu \mathbf{I}), \quad K_{ij} = k(\mathbf{u}_i, \mathbf{u}_j) \tag{8}$$

The hyperparameters $\boldsymbol{\psi} := [\sigma_f, \sigma_\nu, \lambda_1, \ldots, \lambda_{n_\mathbf{u}}]^T$ are commonly unknown and hence need to be inferred from data. In this article the log marginal likelihood $p(\mathbf{y}|\mathbf{U})$ is used. Ignoring constant terms and factors, this can be stated as:

$$\mathcal{L}(\mathcal{D}, \Psi) = -\frac{1}{2}\mathbf{y}^T(\mathbf{K} + \sigma^2_\nu \mathbf{I})^{-1}\mathbf{y} - \frac{1}{2}\ln|\mathbf{K} + \sigma^2_\nu \mathbf{I}|. \tag{9}$$





The required maximum likelihood estimate is then given by $\hat{\boldsymbol{\psi}} \in \arg\max_{\boldsymbol{\psi}} \mathcal{L}(\mathcal{D}, \boldsymbol{\psi})$.

Next we require the predictive distribution of $f(\mathbf{u})$ at an arbitrary input $\mathbf{u}$, which can be found by the conditional distribution of $f(\mathbf{u})$ on the data distribution $p(\mathbf{y}|\mathbf{U})$. From the GP assumption this has a closed-form solution and can be stated as:

$$f(\mathbf{u})|\mathcal{D}, \hat{\boldsymbol{\psi}} \sim \mathcal{N}(\mu_{\mathrm{GP}}(\mathbf{u}; \mathcal{D}, \hat{\boldsymbol{\psi}}), \sigma^2_{\mathrm{GP}}(\mathbf{u}; \mathcal{D}, \hat{\boldsymbol{\psi}})) \tag{10}$$


$$\mu_{\mathrm{GP}}(\mathbf{u}; \mathcal{D}, \hat{\boldsymbol{\psi}}) = \mathbf{k}^T(\mathbf{u})(\mathbf{K} + \sigma^2_\nu \mathbf{I})^{-1}\mathbf{y} \tag{11}$$

$$\sigma^2_{\mathrm{GP}}(\mathbf{u}; \mathcal{D}, \hat{\boldsymbol{\psi}})) = \sigma^2_f - \mathbf{k}^T(\mathbf{u})(\mathbf{K} + \sigma^2_\nu \mathbf{I})^{-1}\mathbf{k}(\mathbf{u}) \tag{12}$$

where $\mu_{\mathrm{GP}}(\mathbf{u}; \mathcal{D}, \hat{\boldsymbol{\psi}})$ can be seen as the GP prediction at $\mathbf{u}$ and $\sigma^2_{\mathrm{GP}}(\mathbf{u}; \mathcal{D}, \hat{\boldsymbol{\psi}})$ as a corresponding measure of uncertainty to this prediction. The GP is a non-parametric model. The training data are explicitly required to construct the predictive distribution. For the above expression a matrix of size $M \times M$ must be inverted, which prohibits large data sets.

## 3 Methodology

### 3.1 Modifier Adaptation with Gaussian processes

The use of GPs in a MA approach to overcome the limitation of estimating the plant gradients was first proposed by de Avila Ferreira et al. (2018). The idea is to replace the zeroth- and first-order modifiers of the cost and constraints in (3) with GP regression terms. Since the wind farms considered in this article do not have inequality constraint functions they are not included in this section. However, inequality constraint functions can be easily incorporated into the method.


The training set of the GP to correct the objective function are the controlled inputs of the approximate model and the plant-model mismatch of the objective function. The new optimization problem of the MA scheme with GP modifiers (MA-GP) is

$$\hat{\mathbf{u}}^*_{k+1} = \arg\min_{\mathbf{u}} \phi(\mathbf{u}, \mathbf{y}(\mathbf{u})) + \mu^{\phi_p - \phi}_{\mathrm{GP}, k}(\mathbf{u}; \mathcal{D}_0, \hat{\boldsymbol{\psi}}_0), \qquad s.t. \quad \mathbf{u} \in \mathcal{U}, \tag{13}$$


where the plant-model mismatch of the cost function is modelled by $\mu^{\phi_p - \phi}_{\mathrm{GP}}$. Similar to the original MA scheme the optimal input of Eq. (13) may be filtered with Eq. (5) to reduce the step-size and help stabilize the MA-GP scheme (del Rio Chanona et al., 2019). The whole MA-GP scheme is presented in Algorithm 1 and Fig. 1.

In Algorithm 1 the hyperparameters are updated if *HypOpt* is true. *HypOpt* is a user-defined condition, which allows to update the hyperparameter. The extrema are to update the hyerparameter each iteration or never. The hyperparameter update


is usually the computational bottle-neck of the MA-GP algorithm. Especially for large data sets it can be expected that the hyperparameter do not change much from one iteration to the next. Therefore, it is reasonable to update the hyperparameters less frequent.

---

[1]The wind farm picture is by Erik Wilde from Berkeley, CA, USA https://www.flickr.com/photos/dret/24110028330/, *Wind turbines in southern California 2016*, https://creativecommons.org/licenses/by-sa/2.0/legalcode



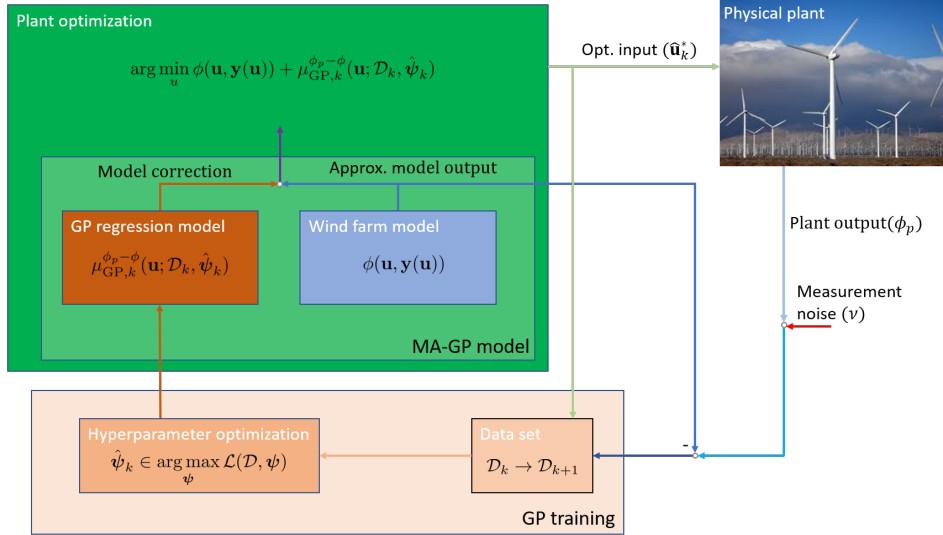

**Figure 1.** The basic idea of the MA-GP scheme for a wind farm. The GP regression model creates an input-output map of the control inputs to the plant-model mismatch. In the MA-GP model the GP regression model is used to correct the output of the approximate model. This MA-GP model is used in the optimization to compute optimal control inputs for the wind farm. The inputs and the difference between the measured and estimated output of plant and model, respectively, are used to update the data set $\mathcal{D}$ and the hyperparameter $\psi$. The measured outputs of the plant are corrupted by noise [1]

---

**Algorithm 1:** Basic MA-GP scheme (del Rio Chanona et al., 2019)

**Initialisation:** GP regression model $\mu_{\mathrm{GP}}^{\phi_p - \phi}$ and hyperparameters $\hat{\psi}_0$ found with (9) and data set $\mathcal{D}_0$; Optimal operation point of the approximate model $\mathbf{u}_0$.

**for** $k = 0, 1, \ldots$ **do**

    Solve modified optimization problem Eq. (13);

    Filter new operating point $\mathbf{u}_{k+1}$ with Eq. (5);

    Evaluate approximate model at new operating point $\mathbf{u}_{k+1}$;

    Obtain measurements of cost function $\phi_p(\mathbf{u}_{k+1})$;

    Update the data set $\mathcal{D}_{k+1}$ with input $\mathbf{u}_{k+1}$ and output $y_{k+1} = \phi_p(\mathbf{u}_{k+1}) - \phi(\mathbf{u}_{k+1}) + \nu_{k+1}.$ ;

    **if** *HypOpt* **then**

        Update hyperparameters $\hat{\psi}_{k+1}$ with new data set $\mathcal{D}_{k+1}$ and Eq. (9) ;

    **end**

    Update GP regression term $\mu_{\mathrm{GP}}^{\phi_p - \phi}$ using $\mathcal{D}_{k+1}$ and hyperparameters $\hat{\psi}_{k+1}$ ;

**end**

---



## 3.2 Numerical turbine and wake models

The wind turbines in the wind farm are represented using the actuator disc theory, which couples the power and thrust coefficient, $C_P$ and $C_T$ (Burton et al., 2011)

$$C_P = 4a(1-a)^2, \tag{14}$$
$$C_T = 4a(1-a), \tag{15}$$

where $a$ is the axial induction factor. The axial induction factor indicates the ratio of wind velocity reduction at the turbine disk compared to the upstream wind velocity. The steady-state power of each turbine under yaw misalignment is given by (Gebraad et al., 2016)

$$P = \frac{1}{2}\rho A C_P \cos\gamma^p u^3, \tag{16}$$

where $A$ is the rotor area, $\rho$ the air density and $p$ a correction factor. In actuator disc theory $p = 3$ (Burton et al., 2011). However, based on large-eddy simulations, the turbine power yaw misalignment has been shown to match the output when $p = 1.88$ for the NREL 5MW turbine (Annoni et al., 2018), which we will use in this article. In the numerical study it will be important to implement a "plant" and model, which are different from each other. Therefore, a second adjusted actuator disk turbine model is created. The FLORIS toolbox (NREL, 2019) contains a table with wind velocities and corresponding thrust and power coefficients of the NREL 5MW turbine. These data are fitted to a new model based on the actuator disk model. The equation for the thrust coefficient $C_T$ is given by Eq. (15) while for the power coefficient $C_P$ three new parameter are identified resulting in

$$C_P = 7.037a(0.625-a)^{1.364}. \tag{17}$$

The model fit is visualised in Fig. 2. Important in the numerical example is the different connection between thrust and power coefficients of both models (Fig. 2b). For the turbine dimensions the NREL 5-MW wind turbine is used (Jonkman et al., 2009). Consequently, the rotor diameter is $D = 136\,\mathrm{m}$ and the hub height $H_H = 90\,\mathrm{m}$.

The Gaussian wake model by Bastankhah and Porté-Agel (2014, 2016) is used to model the flow in the wind farm. The three-dimensional steady-state far wake velocity is assumed to be Gaussian distributed and can be estimated with

$$\frac{\bar{v}(x,y,z)}{\bar{v}_\infty} = 1 - Ce^{-0.5((y-\delta)/\sigma_y)^2}e^{-0.5((z-z_h)/\sigma_z)^2}, \tag{18a}$$

$$C = 1 - \sqrt{1 - \frac{C_T\cos\gamma}{8(\sigma_y\sigma_z/d^2)}}, \tag{18b}$$

where $z_h$ is the tower height, $\delta$ is the wake deflection, and $\sigma_y$ and $\sigma_z$ are the wake widths in lateral and vertical directions. An important variable for the model is the skew angle of the flow past a yawed turbine. The flow skew angle is approximated by

$$\theta \approx \frac{\alpha_1\gamma}{\cos\gamma}\left(1 - \sqrt{1 - C_T\cos\gamma}\right), \tag{19}$$





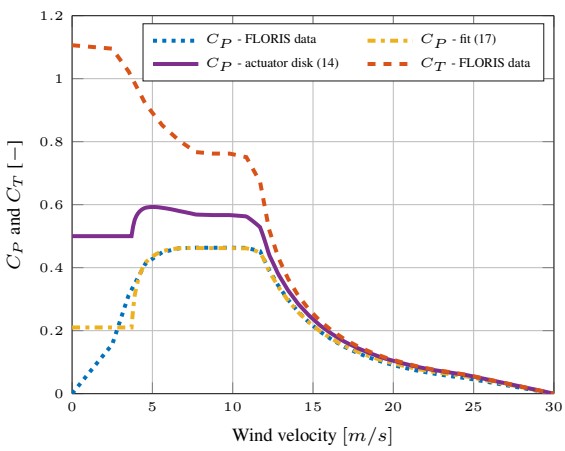

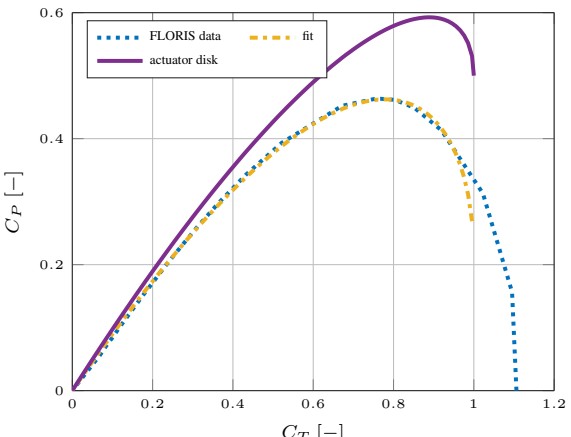

(a) Comparison of $C_P$ values in dependency of the wind velocity. The dashed line gives the corresponding thrust coefficient, which is the same for both models.

(b) $C_P$ - $C_T$ curve of the data and both models.

**Figure 2.** Comparison between data, the new model based on the actuator disk model and the actuator disk model. The thrust coefficients are kept smaller than one for the actuator disk models. The models give a different connection between thrust and power coefficients.

where $\alpha_1$ is a parameter. Bastankhah and Porté-Agel (2016) use $\alpha_1 = 0.3$ and NREL (2019) uses $\alpha_1 = 0.6$ to better fit high-fidelity observations. In the simulation study different values are chosen for this parameter in the plant and approximated model resulting in different optimal operating points.

## 4   Numerical case study

175   In this section numerical results of the MA-GP approach are presented. The control inputs of the wind farms are the yaw angles $\gamma_i$ and the thrust coefficients $C_{T,i}$ of each turbine. Hence, the wind farm has $2N$ control inputs, where $N$ is the amount of wind turbines. The objective of the optimization is to maximize the power production $P_{tot} = \sum_i P_i$ of the wind farm. The control inputs are constrained by box constraints with

$$0 \leq C_{T,i} \leq 0.95, \text{ and } 0° \leq \gamma_i \leq 40°. \tag{20}$$

180   In the MA-GP approach only measurements of the total power output of the wind farm are used. The hyperparameter optimization is performed using the MATLAB optimization toolbox and the nonlinear programming solver *fmincon*. For the optimization of the control inputs of the wind farm the open source software tool *CasADi* (Andersson et al., 2019) is used. *CasADi* is a symbolic framework that provides gradients using Algorithmic Differentiation. The software package *Ipopt* is used as a solver for the nonlinear program (Wächter and Biegler, 2006).



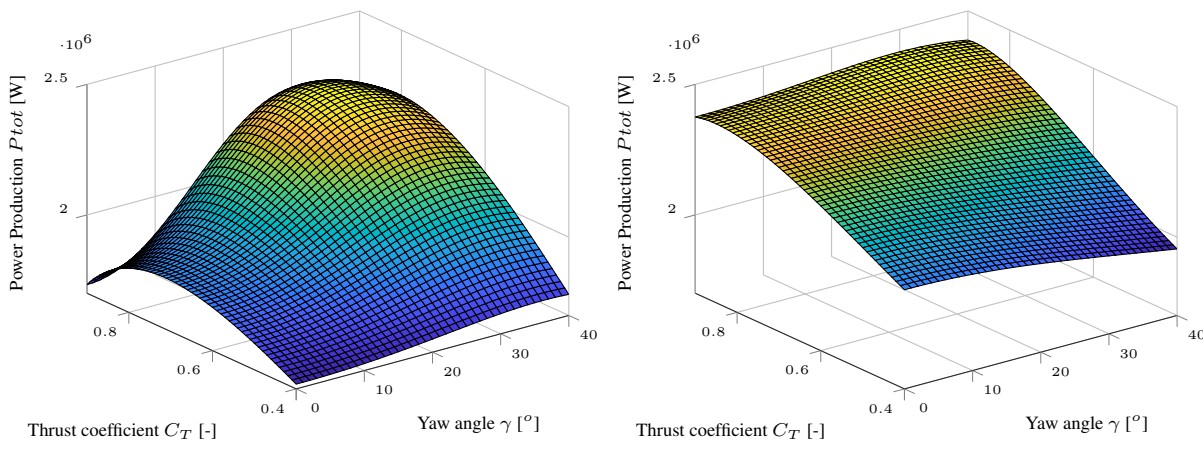

(a) Surface of objective function of plant.    (b) Surface of objective function of approximate model.

**Figure 3.** The power production of plant and approximate model in dependency of the control inputs of the upwind turbine.

### 4.1 Two turbine case

The operating points of two turbines in a row are optimized. The thrust and yaw angle of the downwind turbine are fixed resulting in only two optimization variables in the MA-GP approach. The downwind turbine is operated at its greedy operation point. The turbine row is facing the wind and the spacing between turbines is $5\,\mathrm{D}$. The power production of the wind farm in dependency of the control inputs of the upwind turbine in shown in Fig. 3. The optimal operation point of the plant is $C_{T,p} = 0.82$ and $\gamma_p = 31°$ and of the approximate model $C_{T,p} = 0.89$ and $\gamma_p = 29°$. Indeed, the relative optimization error of the model is only $1.67\,\%$. Still, the model assume that the power production is much less sensitive to changes in the yaw angle, which should be corrected by the MA-GP approach.

Four training points at $C_T = [0.4, 0.8]^T$ and $\gamma = [0°, 25°]^T$ are used to create the initial training set of the GP regression model. The power production of the corrected model in dependency of the control inputs is shown in Fig. 4a. The contour plot of the objective function of the plant, approximate model and MA-GP model after the initial training is shown in Fig. 4b. Clearly four operating points are not sufficient to correct the approximate model correctly. In fact, the optimal operating point of the MA-GP model has an error of $2.87\,\%$, which is larger than the original error of the approximate model.

The MA-GP approach is initialised at the optimal operating point of the approximate model. In each iteration the hyperparameters and the data set of the GP regression model are updated. The new operating point is filtered with Eq. (5) and $\mathbf{L} = \mathrm{diag}(0.4, 0.4)$. The MA-GP approach is able to correct the approximate model and drive the process to its optimal operating point. Fig 5 shows the operating points of the first ten iterations. After four iterations the error in power production is about $0.2\,\%$ and after ten iterations it is $0.0009\,\%$. In addition, the contour lines of the objective function are well approximated. A larger difference between MA-GP model and the plant can be observed at the edges away from the current operating points. Data points at the edges are necessary to improve the identification there. However, to drive the process to its optimal operating points a correct identification of the objective function far away from the maximum is unnecessary. Clearly the initial training

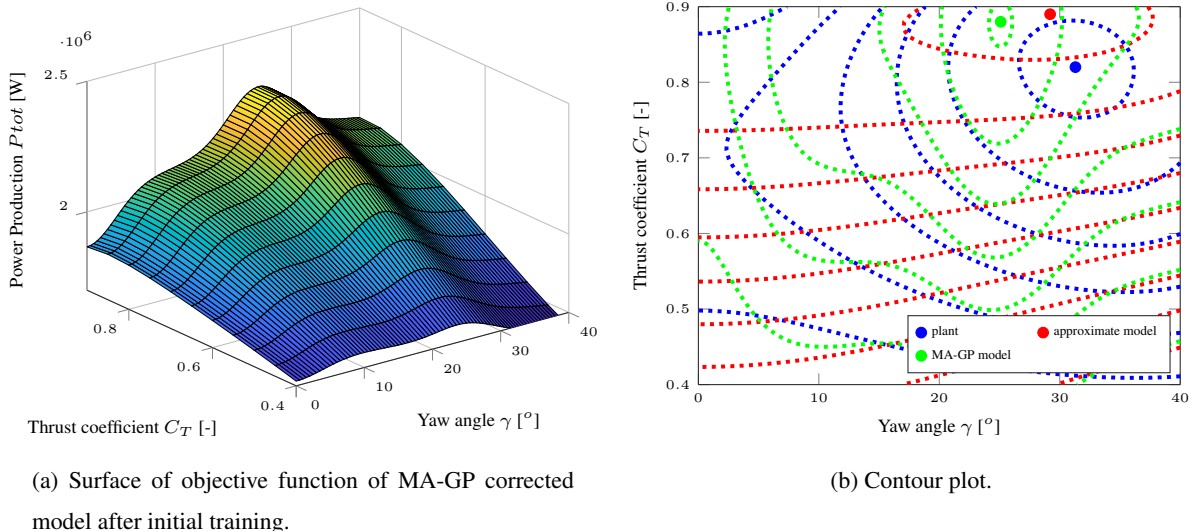

(a) Surface of objective function of MA-GP corrected model after initial training.

(b) Contour plot.

**Figure 4.** The power production of MA-GP model in dependency of the control inputs of the upwind turbine and the contour plot of plant, approximate model and MA-GP model after the initial training.

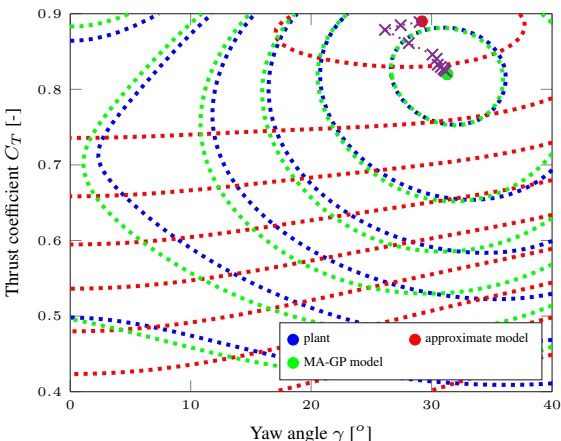

**Figure 5.** The contour plot of plant, approximate model and MA-GP model after ten iterations. The operating points of each iteration are marked with a cross.

set with only four operating points could be increased to improve the identification of the initial model of the MA-GP approach. In the current example it is assumed that the measurements are noise-free. If noise is added to the power measurements the correct identification becomes more challenging and a larger training data set is necessary. A noise with a standard deviation of $50\,\mathrm{kW}$ is added to the measurement. The standard deviation is of the same size as the error in the power production of approximate model and plant at the optimal operating point. A training data set of 20 points is created. After ten iterations the error in the power production is about $0.6\,\%$. The algorithm is able to converge. However, due to the measurement noise a





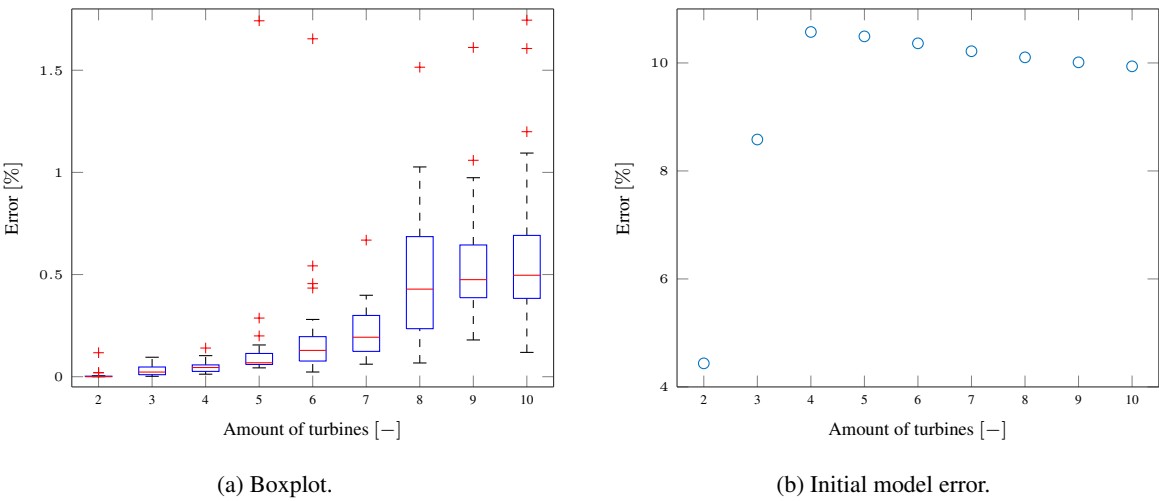

|  (a) Boxplot. | (b) Initial model error. |

**Figure 6.** The boxplot of the optimization results for the differently long wind turbine rows on the left. The red line indicates the median. The bottom and top edges of the blue box indicate the 25th and 75th percentiles, respectively. The red marker indicate outliers and the whiskers extend to the most extreme data points not considered as outliers. The error of the MA-GP approach and the initial error dependent of the amount of turbines in the row. The initial error in the model depending on the amount of turbines in the row on the right.

small error remains after ten iterations. The error can be easily decreased with a larger initial data set, e.g. with a training set of 30 points the error after ten iterations is about $0.35\,\%$.

### 4.2 $n$ turbine row case

In this subsection $n$ turbines aligned in a row are optimized with the MA-GP algorithm. The size of the initial training set is chosen to be $n_d = 10n_u$, where $n_u$ is the amount of control inputs. The operating points of the training set are randomly chosen. The convergence of the MA-GP algorithm is tested on 25 Monte Carlo simulations. The difference between each run is the initial training set.

   The statistic of the error after 25 iterations is shown in Fig. 6a. The error increases with the amount of turbines while it is
almost zero for 2 to 4 turbines. Even though, the error increases with the amount of turbines the algorithm is able to reduce the model error significantly (Fig. 6b). It is not surprising that the error increases with the amount of control inputs. The control inputs are mapped to the total power output of the wind farm. With a large amount of control inputs the correct identification of this input-output map becomes more challenging, which increases the error in the MA-GP algorithm. Again, the error could be decreased with more data in the training set. Currently, the optimization of the process and the optimization of the hyperpa-
rameters takes less than a second even for the ten turbine case. Consequently, it is possible to increase the data set. However, the computational time of the GP regression grows cubic with the amount of data. Therefore, at some point a trade-off between performance and computational time is necessary.

In contrast to purely model-free approaches, e.g. extremum seeking (Johnson and Fritsch, 2012) or MPPT (Gebraad et al.,





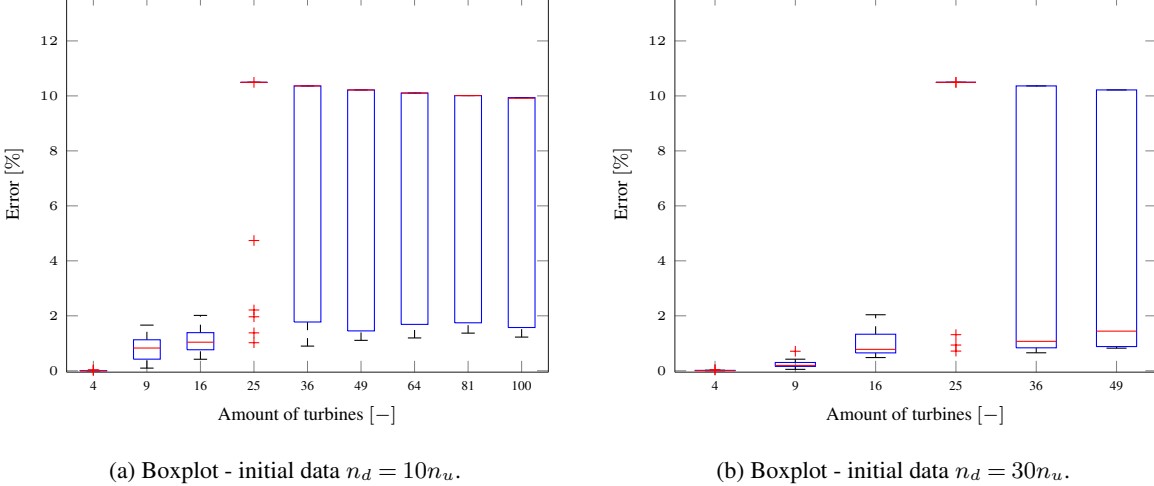

(a) Boxplot - initial data $n_d = 10n_u$.      (b) Boxplot - initial data $n_d = 30n_u$.

**Figure 7.** The boxplot of the optimization results for the differently large wind turbine grids. The red line indicates the median. The bottom and top edges of the blue box indicate the 25th and 75th percentiles, respectively. The red marker indicate outliers and the whiskers extend to the most extreme data points not considered as outliers. The error of the MA-GP approach and the initial error dependent of the amount of turbines in the row. The difference between both runs is the size of the initial training set.

2013), is the MA-GP the algorithm able to find a near optimal point in one iteration. The MA-GP model is already a better rep-
resentation of the plant after the initial training than the approximate model. Nonetheless, measurements close to the optimum can help to refine the MA-GP model.

### 4.3   $n \times n$ turbine grid case

In this subsection the turbines in the wind farm a arranged in a $n \times n$ grid. The wind direction is aligned with the rows of the grid. Interaction between parallel rows is neglectable. Consequently, the wind farm consist of $n$ turbine rows each containing
$n$ turbines. The distance between turbines is $5\,\mathrm{D}$. The identification of the power production of this wind farm layout becomes more challenging. The input space increased and the sensitivity of inputs onto the total power production of the wind farm become similar.

Again the size of the initial training set is chosen to depend linearly on the size of the amount of control inputs with $n_d = 10n_u$. Otherwise the setup is the same as in the turbine row case.
The error after 25 iterations is shown in Fig. 7a. Again the algorithm converges for a small amount of turbines. However, the error in the optimization increases as the amount of turbine increase. Moreover, for grids with 25 and more turbines the majority of the optimizations get stuck at the initial conditions, which is defined by the optimal operation point of the model (Fig. 6b)[2]. Moreover, even in the cases where the MA-GP improves the performance of the wind farm the algorithm converges to errors in the range of $1\,\%$ to $2\,\%$ after 25 iterations. These are much larger than observed in the turbine row case.

---

[2]The percentage in initial error of the turbine row is equal to the percentage in initial error of the grid.





The problems to identify the plant model correctly with a larger inputs space are not surprising. The sample density decreases drastically for larger inputs spaces. The size of the initial training set is increased linearly while it would have to increase exponentially to preserve the same sampling density. For the wind farm with $100$ wind turbines and the current setup the hyperparameter optimization takes usually about $15\,\mathrm{s}$. In some rare cases it took about $5\,\mathrm{min}$. The plant optimization takes less than $10\,\mathrm{s}$. Consequently, the initial data set could be increased to improve the performance of the larger wind farms.

The increase of the initial training set improves the convergence of the method for both small and large inputs spaces (Fig. 7b). Nevertheless, even with the larger size of the initial training set it is challenging to converge to the correct optimum point for cases with a large input space. A larger training set would be necessary for these cases. On the other hand, it also has to be pointed out that the training of the hyperparameters in the GP regression scales cubic with the amount of data. Obviously this limits the size of the initial training set. Otherwise the approach becomes quickly computational infeasible. In

case of an initial set of $n_d = 10n_u$ and a wind farm with $49$ turbines the median time for the hyperparameter optimization is about $3\,\mathrm{s}$. The maximum computational time in the $625$ hyperparameter optimization is about $60\,\mathrm{s}$. In case of an initial set of $n_d = 30n_u$ the median optimization time is about $50\,\mathrm{s}$ while the maximum optimization time is about $23\,\mathrm{min}$. In these cases the optimization algorithm did not converge to an optimum and the maximum amount of iterations until termination was performed. The optimization time could be reduces by limiting the number of iterations. It is expected that it will not influence

the performance since the objective function value in cases the optimization did not converge to an optimum did not change for most of the iterations.

If the MA-GP algorithm for the larger wind farms converges to an optimum it usually takes first a few iterations, where the wind farm is operated at the model optimum point, before the error reduction begins. Obviously the algorithm needs the additional information around the operating point. Interestingly, once the algorithm actually left the initial operating point it converges

relatively quickly to an operating point close to the actual optimum. This is a strong indication that exploration or even just small excitation around an operating point should be activated if the operating point does not change for some time.

Nonetheless, the results show clearly that the MA-GP is able to improve the performance of the model-based optimization for some of the cases. It is not clear how the initial data sets differ for these successful cases. However, it is expected that a large amount of operation points can be excluded from the initial training set of the GP regression since it is known from the

model that they are far away from the optimum operating point. Currently, the initial training set is chosen randomly by Latin hypercube sampling. A smarter selection with a larger density of points around the optimal operating point of the model may improve the MA-GP approach without increasing the initial data set.

## 5   Conclusions

The modifier-adaptation approach with Gaussian processes applied to wind farm control is presented. It is a real-time optimiza-

tion strategy, which corrects optimization model errors by using plant measurements. In the wind farm case the total power production is assumed to be measured and used in the MA-GP approach. The approach works exceptionally well for small input spaces. Here the GP regression is able to correct the model almost perfectly. Consequently, operating points very close




to the real optimum are found in the optimization. For larger input spaces, on the other hand, the error increases. Moreover, for the grid-type wind farm layout with more than 25 turbines convergence with the relatively small initial training sets used in this work could not be achieved at all times.

In future work the performance of the method for large inputs spaces has to be improved. Several ideas are possible to achieve it:

- Increase the training set until it becomes computational unfeasible to increase the training set further.

- Choose the training data points in a smarter way such that they provide enough information about the regions around the expected optimum.

- Extend the algorithm with an exploration part. This can be achieved, for example, by including the variance of the GP regression model in the optimization.

- Include the single turbine power measurements in the identification of the GP regression model. In such a multi-input and multi-output approach the sensitivities of control inputs to the single outputs increase. The model identification should benefit from the approach. Moreover, it is expected that a smaller data set is necessary to achieve the same performance as with the in the article presented multi-inputs and single-output approach. The idea is pursued in Andersson et al. (2020a) with very promising results in increasing the accuracy of the approach with a smaller initial data set.

In addition, the sensitivity of the approach to measurements and inputs noise has to be investigated. In Andersson et al. (2020b) a simple way how to include input noise explicitly in the MA-GP approach is presented. Finally, the model identification should be tested on high fidelity and real data. A preliminary study on a nine turbine wind farm case using data from the high-fidelity simulator SOWFA (Churchfield et al., 2012) will be presented in Andersson et al. (2020c).

*Author contributions.* LEA compiled the literature review, performed numerical simulations, post-processed the data, and wrote the article. LI helped formulate the methodology used in the article and participated in structuring and reviewing of the article.

*Competing interests.* The authors declare that they have no conflict of interest.

*Acknowledgements.* This work was supported by OPWIND, RCN project no. 268044.



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
