# Peer review of "Real-time optimization of wind farms using modifier adaptation and machine learning"

_Wind Energy Science, 2020_

## Referee Comment (RC1) · Bart Doekemeijer (Referee) · 25 Feb 2020

Dear Leif,

Thank you for your article, I read it with much interest. An interesting topic is addressed: the application of machine learning in wind farms to improve surrogate models. I think the work presented itself is of good quality, and needs not much revision. However, I do think notable improvements can be made in how the work is presented. Please consider the following comments:

1. At large, the article seems to be heavily focused on the algorithm, and less so on the application. A large part of the paper is on explaining GPs and MA. Also, the results section contains a fairly large discussion on the number of training points

and the computational cost. Though, what I am really missing is a more in depth discussion on wind farms. How would this algorithm be applied in wind farm control on real sites? What would you consider a training set in real life? Do you need to time averaged measurements? How would you deal with time-varying inflow conditions? Are time delays due to wake propagation an obstacle? You do not have to answer all these questions in this article, but you must acknowledge the relevant challenges for practical application and explicitly state the assumptions in your work. This would also shift the focus in your recommendations section to zoom out more – currently, most recommendations are on improving the algorithm, rather than the application.

2. I believe the article can be significantly reduced without a loss of scientific relevance or clarity. The MA and GP algorithms need not be discussed in detail, since they are presented in the literature elsewhere. The main things you must defend are your assumptions when applying these algorithms. I believe you can condense pages 3-6 into about 1.5 page, presenting the core contribution/application and leading to more clarity and a better focus in the article.

3. The literature survey presented in the introduction presents several high-quality articles. However, the current focus seems to be on general wind farm control. I believe it would be useful to shift the focus to surrogate modeling and model adaptation in wind farms. The article states that the two-step approach leads to suboptimal results. Are there publications that show this? What publications follow the two-step approach? Also, there is some work on using GPs in wind farm control already – it would be useful to cite this work instead.

4. I find it hard to keep oversight and focus when reading the article. Reducing the article's length and clearly stating the paper's contributions in the introduction will help address this. Also, it helps to start each section with one or two sentences relating the upcoming section to the previous section. Moreover, it may help to gather some information in tables, such as in the results section.

Please see my detailed mark-ups in the attached .pdf. Note that I tend to be critical in my comments, and in no way do I mean to devalue your work. The comments are solely made to further improve the scientific relevance of the manuscript.

Please also note the supplement to this comment:
https://www.wind-energ-sci-discuss.net/wes-2020-18/wes-2020-18-RC1-supplement.pdf

**Supplement:**

[revised manuscript text omitted]

---

## Referee Comment (RC2) · Anonymous Referee #2 · 6 Mar 2020

Thank you for this submission.

My main thought is this a good approach to accomodating model mismatch in application of wind farm control. Critically, learning corrections to a starting model sounds appropriate to the problem, versus a completely black box approach. It seems a very good match between a theoretical approach and an application. I look forward to see this idea advanced and tested in dynamic simulations.

Overall Comments and Questions:

1) Can this approach work in a truly dynamic environment? The current paper compares to a static wake model as the plant, but will the work with varying wind directions and wake propagation delay? I assume this is addressed when compared with LES

results in the future paper, but some comments in this line welcome.

2) In the last sections, difficulties are discussed in solving large layouts. Could this problem be addressed by decomposing the large farms into manageable subsets according to wake interactions?

3) Could you speculate how this approach would handle a non-input-output dependency. Here i'm thinking of turbulence. If for example it is not measured at a farm, and yet the wake model depends on it, and it varies let's say day/night, is this manageable? Phrased differently, let's say in the extreme there need to be two models, a stable and unstable atmospheric condition, but stability is not an input, can this be captured? Is there the possibility of modeling hidden confounders?

Specific Comments:

1) References to literature is complete and well done 2) If algorithms are explained in literature, explanation of optimization routines can be condensed

---

## Author Comment (AC1) · 14 Apr 2020

Dear Bart, Thank you for your very detailed review on our paper. I believe including your comments and recommendation will improve the paper considerably. Here I will respond to your four main points. In addition, I answered the questions in your attached pdf. I will attach the pdf to this comment.

Please see below a more detailed response: Point 1: 1. The article seems heavily focused on the algorithm. A more in-depth discussion on the practicality of the algorithm is missing. Questions you raised are: a. How can the algorithm be applied to wind farm control on real sites? b. What would you consider a training set in real life? c. Do you need to time average measurements? d. How would you deal with time-varying

inflow conditions? e. Are time delays due to wake propagation an obstacle?

These are extremely interesting and important questions, which were also raised by the other referee. My plan is to include a discussion section in the revised version of the article to highlight some of these points. We performed also a LES study, which will be presented at the TORQUE 2020. Based on my own expectations and the LES study I can say: The training set would be at least the wind velocity, wind direction, and the power outputs of the turbines. Turbulence intensity could be considered as an input. I would recommend doing a sensitivity study to evaluate how much the variance in the turbulence intensity affects the outputs (it should be considerably larger than the affect of the input noise in the wind measurements). The algorithm needs time averaged measurements. Without an appropriate filter the variance in the data will be large, which will degrade the performance of the learning algorithm. In the LES study we used 5-minutes averaging, which was enough. However, we used quasi-static inflow conditions without wind direction changes. Wind direction changes will degrade the performance. In general, input noise can be to some extend counteracted with more data (assuming the input noise is not biased). I am unsure how the algorithm will react on biased data. If the bias is consistent – e.g. 5 degrees of in all measurements – it should not degrade the algorithm. For a real-life application, it would be necessary to first collect data. The wind direction can be included as an input variable. Otherwise, for many small circle sectors a new model would have to be identified, which could be impractical. I expect, it is necessary to differentiate between atmospheric conditions and identify separate models for each of these conditions. A multi-model approach could be used. Time delays are difficult to handle. Steady-state data is wanted so it is necessary to wait until the first downstream turbine is affected by changes in the upwind turbine. It is difficult to evaluate how much time delays will degrade the performance of the algorithm. In simulation we performed it seems most of the energy transfer in the plant can be captured with the upstream and the next two downstream turbines.

Point 2: 2. The article can be significantly reduced especially the sections about the

MA and GP algorithms can be condensed. I see you point and will condense these sections. I will also try to shift the focus from the algorithms towards the application.

Point 3: 3. The literature survey presented in the introduction seems to focus on general wind farm control. It would be useful to shift to surrogate modelling and adaptation in wind farms. I will try to include these references and refocus the introduction on these topics. The literature on surrogate modeling creating new engineering wake models is quite large. On the other hand, only a few authors considered adaptation of these models for wind farm control. Some considered parameter estimation for their specific wind farm control application. Many used the parameters supplied by the wind farm model (also often estimated from LES data etc.).

Point 4: 4. Shorten the article, clearly state the contribution in the introduction and also start each section with one or two sentences relating the upcoming section to the previous section. In addition, it may be helpful to gather some information in tables. This can be easily included in the revised version of the article. It also is related to point 2.

Again, thank you for all your comments. I will do my best to include them in the revised version.

Please also note the supplement to this comment:
https://www.wind-energ-sci-discuss.net/wes-2020-18/wes-2020-18-AC1-supplement.pdf

**Supplement:**

Dear Bart,

Thank you for your very detailed review on our paper. I believe including your comments and recommendation will improve the paper considerably. Here I will respond to your four main points. In addition, I answered the questions in your attached pdf. This I will attach to this document.

Please see below a more detailed response:

Point 1:

1. The article seems heavily focused on the algorithm. A more in-depth discussion on the practicality of the algorithm is missing. Questions you raised are:
    a. How can the algorithm be applied to wind farm control on real sites?
    b. What would you consider a training set in real life?
    c. Do you need to time average measurements?
    d. How would you deal with time-varying inflow conditions?
    e. Are time delays due to wake propagation an obstacle?

These are extremely interesting and important questions, which were also raised by the other referee. My plan is to include a discussion section in the revised version of the article to highlight some of these points. We performed also a LES study, which will be presented at the TORQUE 2020. Based on my own expectations and the LES study I can say:

The training set would be at least the wind velocity, wind direction, and the power outputs of the turbines. Turbulence intensity could be considered as an input. I would recommend doing a sensitivity study to evaluate how much the variance in the turbulence intensity affects the outputs (it should be considerably larger than the affect of the input noise in the wind measurements).

The algorithm needs time averaged measurements. Without an appropriate filter the variance in the data will be large, which will degrade the performance of the learning algorithm. In the LES study we used 5-minutes averaging, which was enough. However, we used quasi-static inflow conditions without wind direction changes. Wind direction changes will degrade the performance. In general, input noise can be to some extend counteracted with more data (assuming the input noise is not biased). I am unsure how the algorithm will react on biased data. If the bias is consistent – e.g. 5 degrees of in all measurements – it should not degrade the algorithm.

For a real-life application, it would be necessary to first collect data. The wind direction can be included as an input variable. Otherwise, for many small circle sectors a new model would have to be identified, which could be impractical. I expect, it is necessary to differentiate between atmospheric conditions and identify separate models for each of these conditions. A multi-model

approach could be used. Time delays are difficult to handle. Steady-state data is wanted so it is necessary to wait until the first downstream turbine is affected by changes in the upwind turbine. It is difficult to evaluate how much time delays will degrade the performance of the algorithm. In simulation we performed it seems most of the energy transfer in the plant can be captured with the upstream and the next two downstream turbines.

Point 2:

2. The article can be significantly reduced especially the sections about the MA and GP algorithms can be condensed.

I see you point and will condense these sections. I will also try to shift the focus from the algorithms towards the application.

Point 3:

3. The literature survey presented in the introduction seems to focus on general wind farm control. It would be useful to shift to surrogate modelling and adaptation in wind farms.

I will try to include these references and refocus the introduction on these topics. The literature on surrogate modeling creating new engineering wake models is quite large. On the other hand, only a few authors considered adaptation of these models for wind farm control. Some considered parameter estimation for their specific wind farm control application. Many used the parameters supplied by the wind farm model (also often estimated from LES data etc.).

Point 4:

4. Shorten the article, clearly state the contribution in the introduction and also start each section with one or two sentences relating the upcoming section to the previous section. In addition, it may be helpful to gather some information in tables.

This can be easily included in the revised version of the article. It also is related to point 2.

Again, thank you for all your comments. I will do my best to include them in the revised version.

[revised manuscript text omitted]

---

## Author Comment (AC2) · 14 Apr 2020

Thank you for the response to our article. I believe most of the comments can be addressed in a discussion section of the final version of the paper. We investigated some of the questions in a LES study. However, as the nature of these studies the data and test tests are limited by computational constraints. The other referee pointed also out that some parts of the paper can be condensed especially the explanation of the algorithms. I will follow these recommendations in the revised version of the article. I have added my response also in the supplement.

Please see below a more detailed response to your questions:

Questions in comment 1 Can this approach work in truly dynamic environment? Will

the approach work with varying wind directions and wake propagation delay?

These are extremely interesting and important questions. It was also pointed out by the other referee. If the approach would not be applicable to such environments it would be inapplicable to wind farm control. In our LES study that we plan to present at the TORQUE 2020 we simulated a nine-turbine plant with quasi-static wind conditions. The wind direction did not change, but we applied a turbulence intensity of 5%. We filtered the power output with an averaging filter. The approach was able to improve the power production compared to the Gaussian model (with tuned parameter via parameter estimation) about 2-3%. How a complete dynamic case with uncertainties in wind velocity, direction and yaw angle will affect the approach is difficult to say. The approach will require more data to cope with the variance in the training data. The performance will decrease like in robust approaches, which consider uncertainties explicitly. Nevertheless, we expect the approach will still improve the performance of the wind farm.

Questions in comment 2 Could the problem to solve large layouts be addressed by decomposing the large wind farm into manageable subsets according to wake interactions?

One way is to separate the farm into subsets according to wake interactions. Park and Law (2016) proposed such an approach. The other approach is to include the power measurements of each wind turbine in the model identification. Currently, we use a MISO approach approximating the total power production of the plant. A more efficient use of the available measurements is to identify the power production of each turbine and combine these N models in the optimization to optimize the total power output. It is a distributed learning strategy. In simulation studies we were able to show that this distributed (MIMO) approach scales much better for large wind farms. It needs much less data to achieve the same performance as the MISO approach. The distributed learning approach can be combined with the subset approach. Even thought, the GP learning can identify these subsets it can be helpful to specify them

explicitly. The disadvantage of the distributed approach is the requirement to identify N models (which can be parallelized). The disadvantage of the subset approach is the inflexibility it introduces. Depending on the wind directions and the resulting different subset structures x models would have to be identified for each of these structures.

Questions in comment 3 How would the approach handle a non-input-output dependency, like turbulence, which varies on day/night basis? If in the extreme two models for stable and unstable atmospheric conditions are needed, is there a possibility of modeling hidden confounders?

It depends heavily on the influence of the non-measured input to the output. The approach can work without measuring every input. However, if for example turbulence is not explicitly considered in the GP model's inputs its influence will be averaged (over the training data set). In addition, it will increase the variance of the output of the GP. Conditions like stable and unstable atmospheric conditions where the response of the wind farm can differ drastically have to be approached by separate models. If approximated by one model the model will again average the output of these two conditions. This might decrease performance of the control approach. I would propose to differentiate in the data collection of the training data between atmospheric conditions and create several models. It would not be necessary to consider the atmospheric condition as an explicit input to the model. During operations it should be possible to estimate which model is most accurate in the current situation and hence estimate the atmospheric condition. The most accurate model would be used in the optimization. Another way would be a multi-model approach in which each model is weighted: Power $=\phi_1 M_1 +\phi_2 M_2+(1-\sum \phi_i) M_3$. The parameter $\phi_i$ would be estimated using approach proposed in the literature about statistical learning. However, for the multi-model approach I am unsure if an interpolation between models for different atmospheric condition would be appropriate.

Again, thank you for all your comments. I hope I could answer some of your questions. I will try to include some of them in a discussion section at the end of the paper.

Please also note the supplement to this comment:
https://www.wind-energ-sci-discuss.net/wes-2020-18/wes-2020-18-AC2-supplement.pdf

---

## Author Response (AR1)

**Response to Referees for Wind Energy Science submission "Real-time optimization of wind farms using modifier adaptation and machine learning" by Leif Erik Andersson and Lars Imsland**

Dear Referees,

Thank you for the detailed and thorough review of our manuscript "Real-time optimization of wind farms using modifier adaptation and machine learning" by Leif Erik Andersson and Lars Imsland, which we submitted for publication in Wind Energy Science. We appreciate the suggestions and comments to improve the draft. The following changes were made to the manuscript:

- We rewrote the abstract to make it more related to the contribution of the article
- We improved to introduction to shift the focus also on surrogate modelling and adaptation.
- We included a discussion section, which includes some points of our discussion. It gives an outlook on challenges and possible solution to apply the presented method to real wind farms with wake propagation delay, dynamic environment etc.
- We condensed the presentation of the methods, e.g. Gaussian processes and modifier adaptation.
- We rearranged some parts in the presentation of the numerical case study to ease the read and avoid repetition.

These are the main changes in the revised version of our manuscript. Please find below a detailed answer to your comments.

Again, thank you for your time and suggestions to improve our manuscript

On behalf of the authors, yours sincerely,

Leif Erik Andersson

- Attached:
  - Response to Referee #1
  - Response to Referee #2
  - Response to comments in the pdf-file of Referee #1
  - Marked-up manuscript version
  - Marked-up manuscript version (without strikethrough text)

**Response to Referee #1:**
Thank you for your very detailed review on our paper. Including your comments and recommendation improved the paper considerably. Here we will respond to your four main points. In addition, we answered the questions in your attached pdf. This we will attach to this document.

Please see below a more detailed response:

Point 1:

1. The article seems heavily focused on the algorithm. A more in-depth discussion on the practicality of the algorithm is missing. Questions you raised are:
    a. How can the algorithm be applied to wind farm control on real sites?
    b. What would you consider a training set in real life?
    c. Do you need to time average measurements?
    d. How would you deal with time-varying inflow conditions?
    e. Are time delays due to wake propagation an obstacle?

These are extremely interesting and important questions, which were also raised by the other referee. We included a discussion section in the revised version of the article to highlight some of these points. We performed also a LES study, which will be presented at the TORQUE 2020. Based on my own expectations and the LES study I can say:

The training set would be at least the wind velocity, wind direction, and the power outputs of the turbines. Turbulence intensity could be considered as an input. I would recommend doing a sensitivity study to evaluate how much the variance in the turbulence intensity affects the outputs (it should be considerably larger than the effect of the input noise in the wind measurements).

The algorithm needs time averaged measurements. Without an appropriate filter the variance in the data will be large, which will degrade the performance of the learning algorithm. In the LES study we used 5-minutes averaging, which was enough. However, we used quasi-static inflow conditions without wind direction changes. Wind direction changes will degrade the performance. In general, input noise can be to some extend counteracted with more data. I am unsure how the algorithm will react on biased data. If the bias is consistent – e.g. 5 degrees of in all measurements – it should not degrade the algorithm.

For a real-life application, it would be necessary to first collect data – it is possible to create data with high-fidelity models, which would already improve the surrogate model using our adaptation approach. The wind direction can be included as an input variable. Otherwise, for many small circle sectors a new model would have to be identified, which could be impractical. I expect, it is necessary to differentiate between atmospheric conditions and identify separate models for each of these conditions. A multi-model approach could be used. Time delays are difficult to handle. Steady-state

data is wanted so it is necessary to wait until the first downstream turbine is affected by changes in the upwind turbine. It is difficult to evaluate how much time delays will degrade the performance of the algorithm. In simulation we performed it seems most of the energy transfer in the plant can be captured with the upstream and the next two downstream turbines.

Point 2:

2.  The article can be significantly reduced especially the sections about the MA and GP algorithms can be condensed.

We condensed the sections about the MA and GP algorithms. However, the additions of the discussion section did not reduce the overall length of the article.

Point 3:

3.  The literature survey presented in the introduction seems to focus on general wind farm control. It would be useful to shift to surrogate modelling and adaptation in wind farms.

We refocused the introduction. We included surrogate modelling and adaptation algorithms for wind farms.

Point 4:

4.  Shorten the article, clearly state the contribution in the introduction and also start each section with one or two sentences relating the upcoming section to the previous section. In addition, it may be helpful to gather some information in tables.

We followed these recommendations.

**Response to Referee #2:**
Thank you for the response to our article. We investigated some of your questions in a LES study. However, as the nature of these studies the data and test tests are limited by computational constraints. The other referee also pointed also out that some parts of the paper can be condensed especially the explanation of the algorithms. We followed the suggestion in the revised version.

Please see below a more detailed response to your questions:

Questions in comment 1

1. Can this approach work in truly dynamic environment?
2. Will the approach work with varying wind directions and wake propagation delay?

> These are extremely interesting and important questions. It was also pointed out by the other referee. If the approach would not be applicable to such environments it would be inapplicable to wind farm control. In our LES study that we plan to present at the TORQUE 2020 we simulated a nine-turbine plant with quasi-static wind conditions. The wind direction did not change, but we applied a turbulence intensity of 5%. We filtered the power output with an averaging filter. The approach was able to improve the power production compared to the Gaussian model (with tuned parameter via parameter estimation) about 2-3%. How a complete dynamic case with uncertainties in wind velocity, direction and yaw angle will affect the approach is difficult to say. The approach will require more data to cope with the variance in the training data. The performance will decrease like in robust approaches, which consider uncertainties explicitly. Nevertheless, we expect the approach will still improve the performance of the wind farm.

Questions in comment 2

1. Could the problem to solve large layouts be addressed by decomposing the large wind farm into manageable subsets according to wake interactions?

> One way is to separate the farm into subsets according to wake interactions. Park and Law (2016) proposed such an approach. The other approach is to include the power measurements of each wind turbine in the model identification. Currently, we use a MISO approach approximating the total power production of the plant. A more efficient use of the available measurements is to identify the power production of each turbine and combine these N models in the optimization to optimize the total power output. It is a distributed learning strategy. In simulation studies we were able to show that this distributed (MIMO) approach scales much better for large wind farms. It needs much less data to achieve the same performance as the MISO approach. The distributed

learning approach can be combined with the subset approach. Even thought, the GP learning can identify these subsets it can be helpful to specify them explicitly. The disadvantage of the distributed approach is the requirement to identify N models (which can be parallelized). The disadvantage of the subset approach is the inflexibility it introduces. Depending on the wind directions and the resulting different subset structures x models would have to be identified for each of these structures.

Questions in comment 3

1. How would the approach handle a non-input-output dependency, like turbulence, which varies on day/night basis?
2. If in the extreme two models for stable and unstable atmospheric conditions are needed, is there a possibility of modeling hidden confounders?

It depends heavily on the influence of the non-measured input to the output. The approach can work without measuring every input. However, if for example turbulence is not explicitly considered in the GP model's inputs its influence will be averaged (over the training data set). In addition, it will increase the variance of the output of the GP.

Conditions like stable and unstable atmospheric conditions where the response of the wind farm can differ drastically have to be approached by separate models. If approximated by one model the model will again average the output of these two conditions. This might decrease performance of the control approach.

I would propose to differentiate in the data collection of the training data between atmospheric conditions and create several models. It would not be necessary to consider the atmospheric condition as an explicit input to the model. During operations it should be possible to estimate which model is most accurate in the current situation and hence estimate the atmospheric condition. The most accurate model would be used in the optimization. Another way would be a multi-model approach in which each model is weighted: $Power = \phi_1 M_1 + \phi_2 M_2 + (1 - \sum \phi)M_3$. The parameter $\phi_i$ would be estimated using approach proposed in the literature about statistical learning. However, for the multi-model approach I am unsure if an interpolation between models for different atmospheric condition would be appropriate.

©c Author(s) 2020. CC BY 4.0 License.

[revised manuscript text omitted]

[..51]    [..52]

[..53]    [..54]

[..55]
* * *
[36]removed: and Gaussian process regression is explained.

[37]removed: on several examples

[38]removed: The optimization problem of the

[39]removed: plant performance subject to constraints can

[40]removed: (Marchetti et al., 2016):

[46]removed: and $\mathbf{y}_p \in \mathbb{R}^{n_y}$

[47]removed: and output variables, respectively; $\mathbf{u} \in \mathbb{R}^{n_u}$

[48]removed: $\mathbf{y}_p \in \mathbb{R}^{n_y}$ are the input-output pairs of the wind farm; $\phi_p : \mathbb{R}^{n_u} \to \mathbb{R}$ is the cost function to be minimized; $g_{p,j} : \mathbb{R}^{n_u} \times \mathbb{R}^{n_y} \to \mathbb{R}$, $j = 1, \dots, n_g$, are the inequality constraint functions; and

[49]removed: Formulation (1) assumes that $\phi_p$ and $g_{p,j}$ as functions of $\mathbf{u}$, and $\mathbf{y}_p$ are exactly known. However, in any practical application the exact input-output map of the plant is unknown and instead an approximate

[50]removed: system is exploited for the optimization:

80 [..[56] ][..[57] ]plant is available. Consequently, it is not guaranteed that the optimal point of the surrogate model coincide with the optimal point of the plant. MA treats this challenge by directly adapting the optimization problem using plant measurement to allow convergence to the overall plant optimum (Marchetti et al., 2009). The standard MA [..[58] ]adds first order modifiers to correct the gradient of the [..[59] ]

[..[60]]

85 [..[61]] [..[62]]

[..[63]]

[..[64] ]

[..[65]] [..[66]]

[..[67]] [..[68]]

90 [..[69]] [..[70]]

[..[71] ]

[..[72]]

[..[73] ][..[74] ]surrogate model. However, the estimation of the plant gradients [..[75] ]in each iteration is experimentally expensive and the main bottleneck [..[76] ]of the MA implementation in practice (Marchetti et al., 2016). In this article, GPs are used

95 instead to correct the surrogate model (de Avila Ferreira et al., 2018), and by this alleviating the limitation of MA. The next section gives a brief introduction to GPs, before the new optimization problem of the MA-GP approach is stated.

**3 Methodology**

In this section the modifier adaptation approach with Gaussian processes for wind farm control is introduced in Sections 3.1 and 3.1. Thereafter, in Section 3.2, the turbine and wake models used in the case study are explained.
* * *
[56]removed: where the quantities $\phi$, $g_j(\mathbf{u}, \mathbf{y}(\mathbf{u}))$, $\mathbf{u}*$, and $G_j$ refer to the inexact model counterparts of the true plantoptimization problem in Eq. (1).

[57]removed: RTO takes advantage of the available measurements to compensate for plant-model mismatch and adapt the model-based optimization problem Eq. (??) to reach plant optimality.

[58]removed: approach applies first-order correction terms that are added to

[59]removed: cost and constraint functions to match the necessary conditions of optimality upon convergence (Marchetti et al., 2009). Iteratively the following modified optimization problem is solved:

[64]removed: where $\hat{\mathbf{u}}^*_{k+1}$ is the optimal solution at iteration $k+1$, the $\varepsilon_{j,k} \in \mathbb{R}$ are the zeroth-order modifiers for the constraints, and $\boldsymbol{\lambda}^{\phi}_k$ and $\boldsymbol{\lambda}^{G_j}_k$ are the first-order modifiers for the cost and constraints, respectively. The correction terms are given by:

[71]removed: It is recommended to filter the input update $\hat{\mathbf{u}}^*_{k+1}$ to avoid excessive correction and reduce sensitivity to noise (Marchetti et al., 2016):

[73]removed: with $\mathbf{L} = \text{diag}(l_1, \ldots, l_{n_u})$, $l_i \in (0, 1]$ where $l_i$ may be reduced to help stabilize the iterations.

[74]removed: The MA scheme requires the

[75]removed: at each RTO iteration , which

[76]removed: for

**3.1 Gaussian processes**

In this section we give a brief outline of GP regression[..[77] ], for more information [..[78] ]consult Rasmussen and Williams (2006). GP regression [..[79] ]identifies an unknown function $f : \mathbb{R}^{n_u} \rightarrow \mathbb{R}$ from data. [..[80] ]It is assumed that the noisy observations of $f(\cdot)$ [..[81] ]are given by:

$$y_k = f(\mathbf{u}_k) + \nu_k \tag{2}$$

where the value $f(\cdot)$ is perturbed by Gaussian noise $\nu_k$ with zero mean and variance $\sigma_\nu^2$, $\nu_k \sim \mathcal{N}(0, \sigma_\nu^2)$.

[..[82] ]In GP regression, $f(\cdot)$ [..[83] ]is considered a distribution over functions. In this paper, we assume this distribution has a zero mean function and the squared-exponential (SE) covariance function. The choice of the mean and covariance functions assume certain smoothness and continuity properties of the underlying function (Snelson and Ghahramani, 2006), which seems to be a good fit for the plant-model mismatch of the surrogate model. The SE covariance function can be expressed as follows:

$$k(\mathbf{u}_i, \mathbf{u}_j) = \sigma_f^2 \exp\left( -\frac{1}{2} (\mathbf{u}_i - \mathbf{u}_j)^T \mathbf{\Lambda}^{-1} (\mathbf{u}_i - \mathbf{u}_j) \right) \tag{3}$$

where $\sigma_f^2$ is the covariance magnitude and $\mathbf{\Lambda} = \text{diag}(\lambda_1^2, \ldots, \lambda_{n_u}^2)$ is a scaling matrix.

[..[84] ]

[..[85]][..[86]][..[87]]

[..[88] ]

[..[89]]
* * *
[77] removed: for our purposes

[78] removed: refer to

[79] removed: aims to identify

[80] removed: Let the noisy observation

[81] removed: be

[82] removed: We assume

[83] removed: to follow a GP with

[revised manuscript text omitted]